# Do hospitals influence geographic variation in admission for preventable hospitalisation? A data linkage study in New South Wales, Australia

Michael O Falster,[1] Alastair H Leyland,[2] Louisa R Jorm[1]

[1]Centre for Big Data Research in Health, University of New South Wales, Sydney, New South Wales, Australia
[2]MRC/CSO Social and Public Health Sciences Unit, University of Glasgow, Glasgow, UK

**Correspondence to**
Dr Michael O Falster;
m.falster@unsw.edu.au

## ABSTRACT

**Objective** Preventable hospitalisations are used internationally as a performance indicator for primary care, but the influence of other health system factors remains poorly understood. This study investigated between-hospital variation in rates of preventable hospitalisation.

**Setting** Linked health survey and hospital admissions data for a cohort study of 266 826 people aged over 45 years in the state of New South Wales, Australia.

**Method** Between-hospital variation in preventable hospitalisation was quantified using cross-classified multiple-membership multilevel Poisson models, adjusted for personal sociodemographic, health and area-level contextual characteristics. Variation was also explored for two conditions unlikely to be influenced by discretionary admission practice: emergency admissions for acute myocardial infarction (AMI) and hip fracture.

**Results** We found significant between-hospital variation in adjusted rates of preventable hospitalisation, with hospitals varying on average 26% from the state mean. Patients served more by community and multipurpose facilities (smaller facilities primarily in rural areas) had higher rates of preventable hospitalisation. Community hospitals had the greatest between-hospital variation, and included the facilities with the highest rates of preventable hospitalisation. There was comparatively little between-hospital variation in rates of admission for AMI and hip fracture.

**Conclusions** Geographic variation in preventable hospitalisation is determined in part by hospitals, reflecting different roles played by community and multipurpose facilities, compared with major and principal referral hospitals, within the community. Care should be taken when interpreting the indicator simply as a performance measure for primary care.

## INTRODUCTION

Preventable hospitalisations are an intuitive, yet contentious, performance indicator for primary care. Also known as hospital admissions for ambulatory care sensitive conditions, rates of preventable hospitalisations are used in Australia[1 2] and internationally as a measure of hospital use that could potentially be prevented through timely and effective access to primary care. These admissions are estimated to cost over $30 billion dollars annually in the USA,[3] presenting significant potential cost savings to the healthcare system. However, rates of preventable hospitalisation in Australia have not declined, despite accounting for 6% of all hospitalisations and being a national performance indicator for over 10 years.[4]

Health system performance measures should be underpinned by strong evidence that improvements will lead to improvements in health outcomes,[5] and the utility of preventable hospitalisations as a performance measure has been challenged accordingly.[6] Initially developed in the USA where large variations in income, workforce and health insurance coverage result in stark disparities in access to primary care,[7 8] the subsequent adoption of the indicator in various international settings has produced a mixed evidence base, particularly in countries with a universal healthcare system such as Australia,[9] Canada[10 11] and the UK.[6] The utility of the indicator is likely to differ according to the characteristics of the patient population, and the barriers and facilitators to accessing care in the health system.

### Strengths and limitations of this study

► The use of novel cross-classified multiple membership multilevel models makes this the first study on preventable hospitalisations to have modelled each of patient-level, area-level and hospital-level effects.
► The use of a large cohort with detailed survey and linked health data allowed adjustment for a large range of patient confounder.
► We had limited data on hospital characteristics and accessibility of primary care.
► The study population may not be representative of the Australian population, being an older cohort (age 45 and over) with a low response rate.

One health system factor which remains poorly understood is the role of hospitals. Differences in a hospital's propensity to admit patients can arise from physician preferences[12] and in-hospital capacity.[7 13 14] Anecdotal reports from the UK suggest that hospitals play a direct role in choosing to admit patients for observation, such as in regional areas where long travel times and limited clinical support can lead to more cautious admission thresholds.[15] Australia has a vast geography, and in remote areas hospitals and emergency departments may be used as a substitute for general practitioner (GP) care.[16] However, evidence on hospitals' influence on preventable hospitalisations is limited: higher rates have been reported in UK hospitals that convert more emergency department presentations into admissions,[17] and in areas in the US with more hospital beds per capita[18]—although the latter finding has been inconsistent.[19 20]

A better understanding of the role of hospitals would improve our understanding of the limitations of preventable hospitalisations as an indicator of primary care. We sought to quantify between-hospital variation in preventable hospitalisation in New South Wales (NSW), Australia, and assess if this variation differs between categories of hospital facilities.

## METHODS

### Study population

This observational study included participants in The Sax Institute's 45 and Up Study, a prospective cohort of 267 014 residents of NSW, Australia, aged 45 and over.[21] Eligible participants were randomly selected between 2006 and 2009 through the Department of Human Services enrolment database. At study entry participants completed a detailed questionnaire containing information on their health and sociodemographic characteristics, and provided informed consent for long-term follow-up, including linkage with administrative health data sets, and use of their data for research purposes.

For each participant, linked data on hospital admissions (between 2000 and 2011) and deaths (between 2006 and 2011) were obtained from the NSW Admitted Patient Data Collection and the NSW Registry of Births Deaths and Marriages mortality data file, respectively. Data linkage was performed probabilistically by the NSW Centre for Health Record Linkage (http://www.cherel. org.au/). Participants were excluded if they had an unknown age, area of residence, or inconsistent records suggesting incorrect linkage (eg, death before date of study entry).

### Hospitalisations, outcomes and exposures

Hospital outcomes were identified using the linked hospital admissions data, from the time of participants' entry into the study (between 2006 and 2009) until death or the end of linked data (31 December 2011), whichever came first. Hospital admissions were restricted to public hospitals only. Transfers and changes in type of care (eg,

from acute to palliative) within a hospital were considered a continuation of the same episode of care.

Preventable hospitalisations were identified according to the 'selected potentially preventable hospitalisations' performance indicator in the Australian National Healthcare Agreement.[22] The indicator is a composite measure of hospital admissions for 21 conditions, including a selection of chronic conditions (eg, diabetes complications, angina, chronic obstructive pulmonary disease), acute conditions (eg, dehydration and gastroenteritis, pyelonephritis, cellulitis) and vaccine-preventable conditions (eg, influenza and pneumonia). Two additional outcome measures, for which hospital admission was unlikely to be influenced by discretionary patterns of care, were used for comparison: emergency admissions for acute myocardial infarction (AMI) and hip fracture.[14] Hospital diagnosis and procedure codes used to identify outcomes are in online supplementary appendix 1. Sensitivity analyses tested a recently suggested modification to the preventable hospitalisation's indicator, categorising preventable hospitalisations as short (≤2 days length of stay (LOS)) and long (3+ days LOS), on the basis that shorter admissions may be more amenable to primary prevention.[23]

All person-level information was derived from the self-reported survey completed at study entry, including participants' age, sex, education, marital status, annual household income, employment, language spoken at home, health insurance status, level of social support, body mass index, healthy behaviours, multi-morbidity, functional limitation, self-rated health and psychological distress. These variables reflect patients' predisposition and need to use health services, with most previously found to be associated with preventable hospitalisation.[9] All variables were treated as categorical, with missing values as an additional category.[9]

Area-level information was assigned according to the Statistical Local Area (SLA) of patient residence: geographic remoteness using the Accessibility/Remoteness Index of Australia; and the effective supply of full-time workload equivalent (FWE) GPs. FWE GPs were derived from aggregated Medicare claims data,[9 24] as the number of claims for GP services for residents of each SLA, divided by the average number of claims per FWE GP in NSW. Population estimates were used to calculate the density of FWE GPs per 10 000 residents of each SLA, and divided into quintiles.

Hospital category was classified according to hospital peer group, a categorisation used for benchmarking and reporting that groups hospitals by the types of services provided.[25] For this analysis, peer groups were collapsed into six broad categories reflecting major differences in the size, role and location of hospitals: principal (>25 000 acute separations per annum), major metropolitan (10–25 000 acute separations per annum), major non-metropolitan (100 00+ acute separations per annum, in rural areas), district (2–10 000 acute separations per annum), community (<2000 acute separations per annum) and multipurpose (smaller facilities providing

integrated acute health, nursing home, hostel, community health, aged care and non-specialised sub-acute services) (detailed definitions in online supplementary appendix 2). Australia has a vast geography with most high-volume facilities located in metropolitan and inner regional areas. The smaller community and multipurpose facilities provide a mix of acute and sub-acute care, with multipurpose able to provide a range of integrated care services as negotiated between government, health practitioners and the community.

## Statistical methods

Between-hospital variation in admission was analysed using cross-classified multiple membership multilevel Poisson models.[26] All models used number of hospitalisations as the outcome and log of the follow-up time as an offset, so as to model 'rates' of admission, and were adjusted for participants' sociodemographic and health characteristics, geographic remoteness and supply of GP services in their area of residence, so the remaining residual variation was that potentially attributable to hospitals.

Multilevel models allow for variation to be partitioned to various 'levels' for analysis, and these models clustered study participants in both their geographic area of residence (SLA) and all potential hospitals of admission. Because a patient could be admitted to any number of hospitals, this clustering was performed using weighted hospital service area networks of all public hospitals servicing the population.[26] Weighting was determined by patterns of patient flow for all-cause admissions at the level of the postal-area.

From these models, hospital-level incidence rate ratios (IRRs) were derived—the admission rate for the hospital relative to the state average rate, taking into account the factors in the model, as well as the size of the hospital's population.[27] The variation between hospital IRRs was measured using the random intercept variance ($\sigma^2$) from the multilevel model, as well as the average relative deviation (ARD) which quantifies, on average, how much these adjusted hospitalisation rates differ from the statewide adjusted admission rate.[28]

Overall IRRs for hospital types were derived by adding parameters for each hospital type in the model. Given the multiple membership structure, the parameters were calculated as the proportion of hospital services provided by each hospital type in the patient's postal-area. Each parameter was centred on the mean group value, and scaled so a single unit increase represents a 10% increase in service provision. All analyses were performed in SAS V.9.4 and MLwiN V.2.35.

## Patient and public involvement

Participants in the 45 and Up Study completed a baseline questionnaire and have provided informed consent for the use of their data for research purposes. However, patients and the public were not involved in the design of this study.

## RESULTS

Of 267 014 participants in the linked dataset, n=119 were excluded because they had unknown area of residence or incompatible dates in the linked data. Participants in 16 postal areas did not have any hospitalisations during follow-up; the 69 participants residing in these areas were excluded, leaving 266 826 for analysis, over an average follow-up of 3.7 years. Mean age, self-reported health and multi-morbidity of study participants were broadly consistent across remoteness categories (table 1), although participants in remote areas were slightly younger, with poorer health and a higher number of comorbidities. Patients were admitted to a total of 259 different facilities, including n=17 principal referral, n=12 major metropolitan, n=12 major non-metropolitan, n=38 district, n=70 community and n=110 multi-disciplinary facilities.

The majority of the 30 264 preventable hospitalisations during follow-up were to principal hospitals (31%) with only a small proportion to community (9.1%) and multipurpose (2.6%) facilities (table 1). However, this pattern was inverted for participants in remote and outer regional areas, with the majority of admissions to community (24.6%) and district hospitals (37.4%). A similar pattern was observed in the 3167 emergency AMI and 1550 emergency hip fracture admissions, although with a smaller proportion of admissions overall to district, community and multipurpose hospitals (data not shown).

There was significant between-hospital variation in preventable hospitalisation, such that each hospital deviated on average 26% from the mean adjusted rate of admission ($\sigma^2$=0.312; SE=0.059; ARD=25.6). This variation was much less pronounced for emergency admissions for AMI ($\sigma^2$=0.047; SE=0.026; ARD=9.6) and was not significant for hip fracture ($\sigma^2$=0.015; SE=0.017; ARD=2.9).

Figure 1 shows hospital-level IRRs from the multilevel model, which indicate how each hospital differs from the state average, after adjusting for patient and geographic factors. There was considerable variation in preventable hospitalisation, with 7% of hospitals having significantly higher or lower than average adjusted rates of admission. When stratified by category of hospital, the greatest variation was seen in community and district hospitals, with community hospitals in particular having the highest rates of preventable hospitalisation—up to four times the average rate of admission. There were no hospitals with significant deviations from the mean for emergency AMI or hip fracture admissions.

ARDs stratified by hospital category (figure 1) corroborated these results, with community hospitals having the highest levels of variation in preventable hospitalisation (average 36% difference from the mean), and principal hospitals varying the least (average 21% difference from the mean). There was less variation between all hospital types for emergency AMI or hip fracture admissions than preventable hospitalisations.

The inclusion of hospital category in the regression models (table 2) showed significantly higher rates of preventable hospitalisations among people serviced by

**Table 1** Cohort characteristics at baseline, and number of preventable hospitalisations during follow-up, by remoteness of area of residence

| | | By remoteness category of residence | | | |
| | **Total** | **Major cities** | **Inner regional** | **Outer regional** | **Remote** |
| --- | --- | --- | --- | --- | --- |
| Cohort characteristics | | | | | |
| N | 266 826 | 119 496 | 94 568 | 47 438 | 5324 |
| Age (mean) | 62.7 | 63.4 | 62.4 | 62.2 | 60.7 |
| Age (IQR) | 53.6–70.4 | 53.6–71.9 | 53.8–69.7 | 53.7–69.4 | 52.0–67.8 |
| Female (%) | 53.6 | 52.4 | 54.7 | 54.3 | 55.5 |
| Fair/poor self-rated health (%) | 13.7 | 13.9 | 13.4 | 13.7 | 16.1 |
| With>3 comorbidities (%) | 7.4 | 7.3 | 7.5 | 7.2 | 8.0 |
| Preventable hospitalisations | | | | | |
| Number of admissions | 30 264 | 12 512 | 10 161 | 6512 | 1079 |
| Admissions to hospital type (%) | | | | | |
| Principal | 9398 (31.0) | 7506 (60.0) | 1600 (15.7) | 255 (3.9) | 37 (3.4) |
| Major metropolitan | 4172 (13.8) | 3321 (26.5) | 787 (7.7) | 61 (0.9) | 3 (0.3) |
| Major non-metropolitan | 6443 (21.3) | 560 (4.5) | 3933 (38.7) | 1872 (28.7) | 78 (7.2) |
| District | 6715 (22.2) | 804 (6.4) | 3070 (30.2) | 2468 (37.9) | 373 (34.6) |
| Community | 2760 (9.1) | 278 (2.2) | 611 (6.1) | 1491 (22.9) | 380 (35.2) |
| Multipurpose | 776 (2.6) | 43 (0.3) | 160 (1.6) | 365 (5.6) | 208 (19.3) |

community (IRR: 1.06; 95% CI 1.02% to 1.10%) and multipurpose (IRR: 1.05; 95% CI 1.01% to 1.09%) than principal hospitals. For emergency AMI admissions, there were significantly higher rates in people serviced by major non-metropolitan (IRR: 1.04; 95% CI 1.02% to 1.07%), and lower rates among people serviced by multipurpose facilities (IRR: 0.93; 95% CI 0.88% to 0.99%). IRRs for all variables in the model are provided in online supplementary appendix 3.

A sensitivity analysis categorising LOS (table 3) found the majority of preventable hospitalisations (n=16 305, 53.9%) were short stay admissions (0–2 days LOS), with the remainder (n=13 959, 46.1%) having a LOS of 3 days or more. There were differing patterns of variation by LOS, with the significantly higher rates of admission for community and multipurpose hospitals restricted to short-stay preventable hospitalisations only.

## DISCUSSION

We found significant variation in rates of preventable hospitalisation between public hospitals, even after adjustment for patient and geographic factors. Our finding was most marked for community and multipurpose hospitals—smaller facilities which provide the majority of services to patients living in regional and remote communities. Given similar variation was not observed for other less-discretionary conditions, major hospitals servicing regional areas, or for admissions with a longer LOS, our findings indicate a varying propensity to admit patients for preventable hospitalisation among and between categories of hospital facilities.

Our findings do not suggest that preventable hospitalisations should be used as indicator of discretionary admission practice – the effect size was modest and, consistent with prior research, the strongest predictors of admission were patient sociodemographic and health characteristics.[9] But while admissions to community and multipurpose hospitals represented only a small proportion (12%) of all preventable hospitalisations, they made up 55% of admissions in remote areas of Australia, where there is both high variability—with over a fivefold variation in rates of preventable hospitalisations[2]—and also the highest rates of admission.[1 2] Accordingly, these differences in admission practices are likely to play an important role in driving geographic variation in the preventable hospitalisations performance indicator. The implications for performance measurement are clear: interpretation of the indicator is complex and factors along the care continuum, including hospitals' propensity to admit, influence variation in admission rates.

There is very little existing evidence about how admissions for preventable hospitalisations vary between hospitals in Australia. One study of major hospitals in NSW reported up to 11-fold and sevenfold variation between hospitals in the proportion of admissions that were for congestive heart failure and chronic obstructive pulmonary disease, respectively,[29] and earlier work from the current team found no association between preventable hospitalisations and hospital bed occupancy rates.[26] Importantly, these previous analyses (as with most hospital reporting) excluded community and multipurpose hospitals—the facilities in this study with the strongest patterns

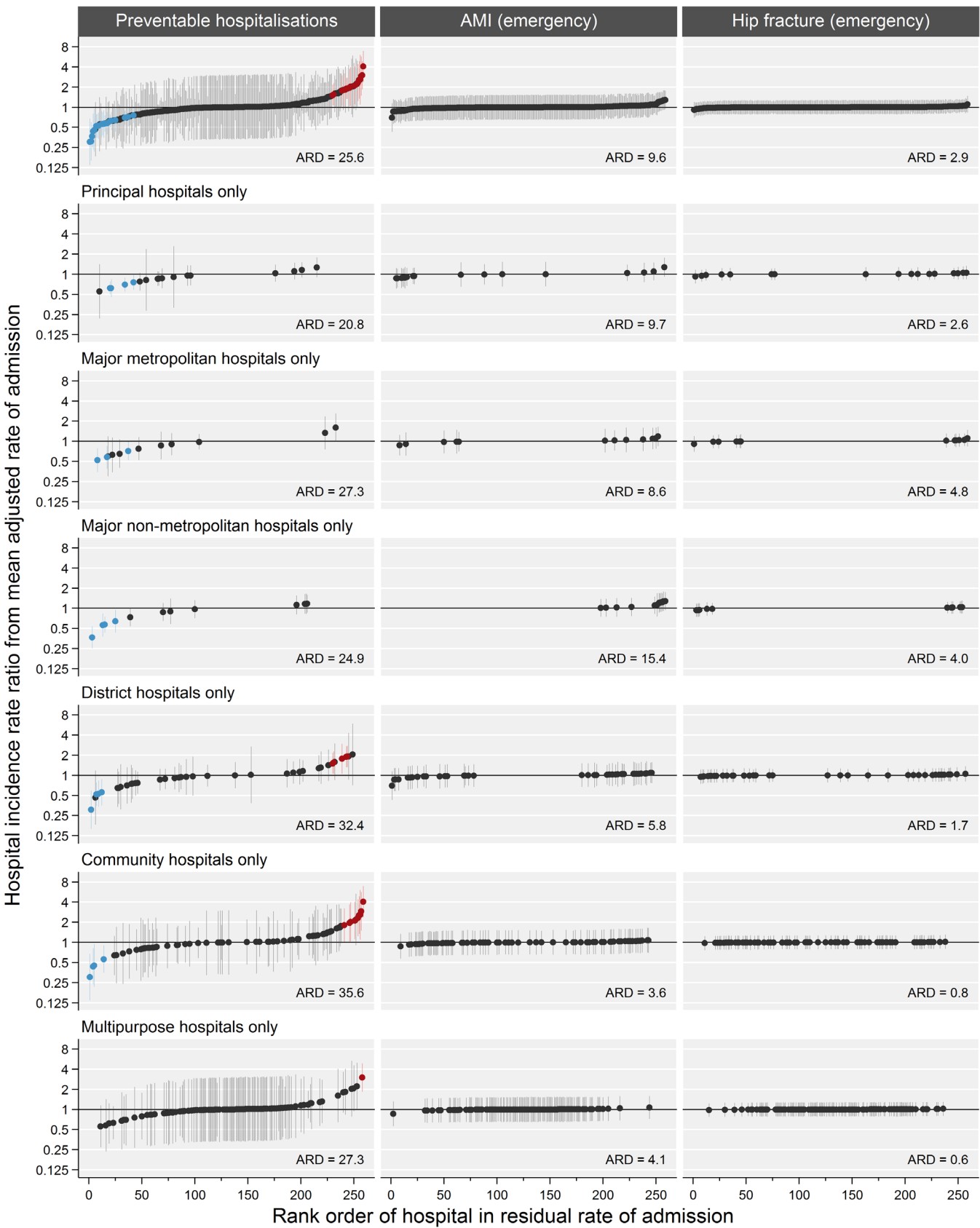

**Figure 1** Hospital-specific incidence rate ratios from the mean adjusted rate of admission, for preventable hospitalisation and emergency admissions for acute myocardial infarction (AMI) and hip fracture, overall and stratified by hospital category. ARD, average relative deviation. Red and blue markers indicate hospitals with significantly higher and lower rates of admission, respectively. Adjusted for patient sociodemographic and health factors, remoteness and supply of general practitioner services in area of residence.

**Table 2** Incidence rate ratio (IRR) of hospital category for preventable hospitalisation and emergency admissions for acute myocardial infarction (AMI) and hip fracture

| Hospital category | Preventable hospitalisations IRR (95% CIs) | AMI (emergency) IRR (95% CIs) | Hip fracture (emergency) IRR (95% CIs) |
|---|---|---|---|
| Principal | 1.00 (ref) | 1.00 (ref) | 1.00 (ref) |
| Major metropolitan | 0.99 (0.95% to 1.03%) | 1.02 (0.99% to 1.05%) | 1.02 (0.99% to 1.05%) |
| Major non-metropolitan | 1.01 (0.97% to 1.04%) | 1.04 (1.02% to 1.07%) | 0.99 (0.96% to 1.02%) |
| District | 1.02 (0.99% to 1.06%) | 1.00 (0.97% to 1.03%) | 0.99 (0.96% to 1.02%) |
| Community | 1.06 (1.02% to 1.10%) | 0.97 (0.93% to 1.01%) | 0.96 (0.91% to 1.01%) |
| Multipurpose | 1.05 (1.01% to 1.09%) | 0.93 (0.88% to 0.99%) | 1.02 (0.94% to 1.09%) |

of variation. It is difficult to assess causes of between-hospital variation in the context of this analysis. Both differences in hospital roles (eg, provision of both acute and sub-acute services) and differences in discretionary admission thresholds (eg, admitting patients for observation to avoid long travel times)[15] could contribute, as well as the provision of community-based services such as hospital in the home.[30]

The preventable hospitalisations indicator is considered a measure of timely and effective access to primary care, and our findings are not inconsistent with this interpretation. Some of the variation in community and multipurpose hospitals is likely to reflect the facility acting as a substitute for primary care in areas where access is poor, and may arguably reflect either a deficiency of primary care or appropriate integration of services to meet population needs. We were unable to examine further dimensions of access, such as waiting times, distance to nearest GP clinic and type of in-hospital practitioner, so were unable to further tease out these effects. However, our results do suggest that use of the preventable hospitalisations indicator beyond its original intent—as a yardstick measure of health system performance[7]—needs to be approached with caution.

Our study is among a few internationally to provide evidence of a hospital-level difference in propensity to admit patients for preventable hospitalisations,[17 18] and is the first to quantify the extent of this variation. The findings, while not directly applicable to different healthcare settings, highlight the contextual differences between health systems which should be considered when adopting international performance indicators, as well as the need for localised policy responses tailored to models of care.

The key strength of this study is the use of a large cohort with detailed survey and linked health data. Much inference on preventable hospitalisation is limited either by unmeasured confounders or the use of ecological measures of patient demographics, and estimation of hospital effects can be difficult given the lack of a discrete population denominator. The use of cross-classified multiple membership multilevel models makes this the only study to perform appropriate modelling for each of patient-, area- and hospital-level effects. A limitation is that unexplained hospital variation remained, and we had only limited data on hospital characteristics, so the impact of more complex models of care, such as integrated care programmes, has yet to be explored. The use of a population cohort meant further measures of morbidity derived from hospital admissions data (eg, Charlson index) were not able to be utilised. Generalizability of our findings may also be limited given the older age (45 years and over) and low response rate (18%) of the study cohort, although the considerable size and heterogeneity of the study mean inferences from within-cohort comparisons remain valid.[31]

## CONCLUSION

Geographic variation in rates of preventable hospitalisation is determined in part by the hospitals themselves,

**Table 3** Average relative deviation (ARD) and incidence rate ratio (IRR) by hospital category for rates of preventable hospitalisation, separated as short-stay (0–2 days length of stay (LOS)) and long-stay (>2 days LOS) admissions

| Hospital category | Short stay (0–2 days LOS) ARD | Short stay (0–2 days LOS) IRR (95% CIs) | Long stay (>2 days LOS) ARD | Long stay (>2 days LOS) IRR (95% CIs) |
|---|---|---|---|---|
| Principal | 17.9 | 1.00 (ref) | 14.6 | 1.00 (ref) |
| Major metropolitan | 25.5 | 0.99 (0.95% to 1.02%) | 25.9 | 1.00 (0.97% to 1.03%) |
| Major non-metropolitan | 22.7 | 1.02 (0.98% to 1.05%) | 11.3 | 0.99 (0.96% to 1.02%) |
| District | 30.4 | 1.02 (0.99% to 1.05%) | 24.3 | 0.98 (0.95% to 1.00%) |
| Community | 17.5 | 1.04 (1.01% to 1.07%) | 25.7 | 1.02 (0.99% to 1.05%) |
| Multipurpose | 24.3 | 1.04 (1.00% to 1.08%) | 11.6 | 0.99 (0.95% to 1.03%) |

reflecting different roles of smaller and rural hospitals compared with major and principal referral hospitals to meet the needs of the community. International adoption of the preventable hospitalisations health performance indicator should consider the contextual barriers and facilitators to accessing care in the relevant health system. In Australia, care should be taken when interpreting preventable hospitalisations simply as a measure of accessibility and quality of primary care.

**Acknowledgements** The authors would like to thank Federico Girosi and the Australian Primary Health Care Research Institute at the Australian National University for assistance with the data. This research was completed using data collected through the 45 and Up Study (www.saxinstitute.org.au). The 45 and Up Study is managed by the Sax Institute in collaboration with major partner Cancer Council NSW and partners: National Heart Foundation of Australia (NSW Division); NSW Ministry of Health; NSW Government Family & Community Services—Ageing, Carers and the Disability Council NSW; and the Australian Red Cross Blood Service. We thank the many thousands of people participating in the 45 and Up Study. We thank the Sax Institute, the NSW Ministry of Health and the NSW Register of Births, Deaths and Marriages for allowing access to the data, and the Centre for Health Record Linkage for conducting the probabilistic linkage of records.

**Contributors** MOF conceived the project, undertook the literature review, performed data analysis and drafted the manuscript. LRJ and AHL provided guidance and interpretation. All three authors edited, reviewed and approved the final manuscript. LRJ conceived the APHID study.

**Funding** This study was funded by a National Health and Medical Research Council Partnership Project Grant (#1036858) and by partner agencies the Australian Commission on Safety and Quality in Health Care, the Agency for Clinical Innovation and the NSW Bureau of Health Information. MF is supported by a National Health and Medical Research Council Early Career Fellowship (#1139133). AL receives funding from the Medical Research Council (MC_UU_12017/13) and the Scottish Government Chief Scientist Office (SPHSU13).

**Competing interests** None declared.

**Patient consent for publication** Not required.

**Ethics approval** Ethics approval for the 45 and Up Study was given by the University of New South Wales Human Research Ethics Committee, and ethics approval for this study was given by the NSW Population and Health Services Research Ethics Committee and the University of Western Sydney Research Ethics Committee. All analyses were carried out in accordance with these approvals.

**Provenance and peer review** Not commissioned; externally peer reviewed.

**Data sharing statement** The data set used for this study was constructed from pre-existing source data sets (routinely collected data and the 45 and Up Study) with the permission from the custodians of each of these data sets and with specific ethical approval. The data set could potentially be made available to other researchers if they obtain the necessary approvals. Further information on this process can be obtained from the 45 and Up Study (45andUp.research@saxinstitute.org.au) and the NSW Centre for Health Record Linkage (cherel.mail@moh.health.nsw.gov.au).

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
