## [Reviewer comments · BMJ Open]

ARTICLE DETAILS

TITLE (PROVISIONAL)	Do hospitals influence geographic variation in admission for preventable hospitalisation? A data linkage study in New South Wales, Australia
AUTHORS	Falster, Michael; Leyland, Alastair; Jorm, Louisa

VERSION 1 – REVIEW

REVIEWER	Reviewer name: Matthew Hensley Institution and Country: University of Michigan, United States Competing interests: None
REVIEW RETURNED	20-Nov-2018

GENERAL COMMENTS	Falster et al. conducted an investigation testing whether there is between-hospital variation in preventable hospitalizations using a multi-level Poisson model. This observational study analyzed a group of patients >45 years in New South Wales, Australia. Overall the methods were thoughtful and statistical methods appropriate. In the methods and results section, I would suggest using an index such as Charlson or Elixhauser (PMID:26351192) to fully assess comorbidities and their contribution to differences between hospital admissions. These indices are generic and used in critically ill patients, so an alternative index used in Australian population could be utilized (PMID: 30326898). Accounting for individual comorbidities such as CHF, malignancy, and cerebrovascular disease may change the results of the model rather than adjusting for number of comorbidities. Overall, the investigation by Falser and colleagues answers an important question regarding variation between hospitals with regards to preventable admissions. The effect size is small but impact on a population basis is important and will assist in developing future health policies.
---

REVIEWER	Reviewer name: Laura Coots Daras, PhD Institution and Country: RTI International Competing interests: None declared
REVIEW RETURNED	26-Nov-2018

GENERAL COMMENTS	1) In the methods section, more information (perhaps a few sentences) about the selected potentially preventable hospitalisations indicator would be helpful for readers that are not familiar with this indicator. 2) It would be helpful if the authors could describe their rationale for using a count of hospitalizations. I would imagine patients with
---

	>1 hospitalization would differ than those with only 1 during the period. Other approaches would be to model this as 0/1 and ignore or exclude subsequent hospitalizations for patients. At a minimum, it would be helpful to know how often in the sample there is >1 hospitalization--perhaps I missed it. 3) Without being familiar with prior debates about the use of this indicator in Australia, the conclusion as summarized in the main paper appears to be a dramatic one. As a reader, I was not convinced by this one analysis that the indicator should not be used or has no value. In the US, we consider these measures as relating to not only quality/access to outpatient care, but also towards understanding care transitions and discharge planning. Is the concern over which provider type should be attributed to these outcomes? Is the indicator framed as only for quality of primary care? Is there an alternative approach or another indicator that would better reflect quality? Perhaps this could be considered more of a health system indicator. A bit more discussion would help. My interpretation is that there should be variation in quality indicators otherwise there is no performance gap or area to target quality improvement. With that said, I really liked that the authors chose two emergent conditions (AMI and hip fx) for comparison purposes. If the authors chose to make this final conclusion about the indicator, I think more information to substantiate the argument is warranted. Otherwise, their statements could be softened.
--	--

REVIEWER	Reviewer name: Ester Angulo-Pueyo Institution and Country: Aragon Institute of Health Sciences (IACS), Spain Competing interests: None declared
REVIEW RETURNED	07-Dec-2018

GENERAL COMMENTS	I would thank for the opportunity to review this interesting study which aims to provide new evidence to refute the idea that preventable hospitalisations (PH) can be used as a mere indicator for primary care performance. At this respect, there is published evidence showing that PH might be associated with different factors, some of them related to primary (ambulatory) care, such as, effective access to healthcare facilities, care continuity across levels or availability of primary care professionals; whereas others are non-ambulatory care related, for example socioeconomic variables, existence of long-term care services, and some hospital related factors such as supply of acute beds or propensity to hospitalisation. In this work, authors specifically focus on the potential influence of hospitals in PH by means of quantifying variation in preventable hospitalisations between hospitals in order to better understand the role of those in that type of admissions. The work is well structured and, in general, clearly explained, resulting in a smooth and interesting reading. My major objection is about the conclusion that authors inferred from their analysis To get their objective, authors include different categories of hospitals in the study, some of them admitting non acute patients. I consider that the purpose and population served by these centres should be detailed (apart from appendix 2) to ease the understanding by non-Australian readers.
---

	Reading the text, I wonder if some community and multipurpose centres that admit non-acute patients could have been specifically devised to admit and treat patients with the conditions considered avoidable because of the lack of ambulatory facilities in the area where they are placed. In this line, supply of GP services is the only variable used referring to primary care and maybe not sufficient to cover all the potential influence of primary (or ambulatory) care in avoidable hospitalisations. Maybe difficulties to access primary care centres or the lack of ambulatory facilities that assure care continuity, are affecting the propensity detected. Therefore, it could not be discarded the effect of inappropriate ambulatory care in the higher risks of admissions detected in some hospitals. If possible, other variables referring to accessibility or utilisation (distance to primary care centres) of primary/ambulatory care should be included in the analysis, or if it is not possible, it should be addressed in discussion/limitations. It has to be said that authors qualify the results in the discussion (lines 26-30), but in conclusions they stated that PH are determined in part by hospitals, despite the fact that according to the results, the differences in admitting patients between hospitals may be a consequence of the lack of accessibility to appropriate care in some areas. I consider this work will benefit from considering additional variables related to primary/ambulatory care and/or further discussion about the interpretation, implications and conclusions derived from the results. Minor things. In results, authors stated “When stratified by category of hospital, the greatest variation was seen in community, district and multipurpose hospitals, [...]”. But in figure 1, only one multipurpose hospital has a significantly different risk from the mean and the ARD of this group (27.3) is similar to the ARD calculated for major metropolitan hospitals, so the sentence should be modified accordingly. Finally, there are some points in Methods section that will benefit from further explanation:  - A definition of full time workload equivalent general practitioners - The calculation of overall IRRs for hospital (current explanation “overall IRRs for hospital types were derived by including the hospital category in the model, as a 10% increase in provision of hospital services to the patient’s postal area, centred on the mean group value” is not explanatory enough), - Detailing the number of hospitals included in each category and the number of short-stay and long-stay admissions included in the sensitive analysis.
--	--

VERSION 1 – AUTHOR RESPONSE

Reviewer: 1

Falster et al. conducted an investigation testing whether there is between-hospital variation in preventable hospitalizations using a multi-level Poisson model. This observational study analyzed a group of patients >45 years in New South Wales, Australia. Overall the methods were thoughtful and statistical methods appropriate.

Comment 3: In the methods and results section, I would suggest using an index such as Charlson or Elixhauser (PMID:26351192) to fully assess comorbidities and their contribution to differences between hospital admissions. These indices are generic and used in critically ill patients, so an alternative index used in Australian population could be utilized (PMID: 30326898). Accounting for individual comorbidities such as CHF, malignancy, and cerebrovascular disease may change the results of the model rather than adjusting for number of comorbidities.

Overall, the investigation by Falser and colleagues answers an important question regarding variation between hospitals with regards to preventable admissions. The effect size is small but impact on a population basis is important and will assist in developing future health policies.

Response: Thank you for your feedback. While we agree morbidity scores such as the Charlson or Elixhauser Index can be useful for further assessing patient's comorbid conditions, the population cohort design of the study limits application in this analysis. The majority of patients in our study were not admitted to hospital, and so a comorbidity score derived from hospital admission data could only be calculated for a limited subgroup of patients and would therefore introduce a new potential source of bias. We have briefly acknowledged this in the limitations (page 8).

"The use of a population cohort meant further measures of morbidity derived from hospital admissions data (e.g. Charlson index) were not able to be utilised."

Reviewer: 2

Comment 4: In the methods section, more information (perhaps a few sentences) about the selected potentially preventable hospitalisations indicator would be helpful for readers that are not familiar with this indicator.

Response: We have added further information about the potentially preventable hospitalisations indicator (page 4).

"Preventable hospitalisations were identified according to the 'selected potentially preventable hospitalisations' performance indicator in the Australian National Healthcare Agreement.²² The indicator is a composite measure of hospital admissions for 21 conditions, including a selection of chronic conditions (e.g. diabetes complications, angina, chronic obstructive pulmonary disease), acute conditions (e.g. dehydration and gastroenteritis, pyelonephritis, cellulitis) and vaccine-preventable conditions (e.g. influenza and pneumonia)."

Comment 5: It would be helpful if the authors could describe their rationale for using a count of hospitalizations. I would imagine patients with >1 hospitalization would differ than those with only 1 during the period. Other approaches would be to model this as 0/1 and ignore or exclude subsequent hospitalizations for patients. At a minimum, it would be helpful to know how often in the sample there is >1 hospitalization--perhaps I missed it.

Response: Our motivation for modelling counts instead of a binomial outcome (i.e. ever/never hospitalised) was that (a) we believe patients with >1 admission are different to those who have just 1 admission, as the reviewer suggests, and; (b) patients in our study cohort had variable lengths of follow-up time, and we wished to maximise the utility of the data. We note this modelling approach is regularly used in the analysis and reporting on preventable hospitalisation.

We have previously reported on the number of preventable hospitalisations per patient within this study cohort, including breakdown by type of condition, and prefer not to repeat this information (see <https://doi.org/10.1097/MLR.0000000000000342>).

Comment 6: Without being familiar with prior debates about the use of this indicator in Australia, the conclusion as summarized in the main paper appears to be a dramatic one. As a reader, I was not convinced by this one analysis that the indicator should not be used or has no value. In the US, we consider these measures as relating to not only quality/access to outpatient care, but also towards understanding care transitions and discharge planning. Is the concern over which provider type should be attributed to these outcomes? Is the indicator framed as only for quality of primary care? Is there an alternative approach or another indicator that would better reflect quality? Perhaps this could be considered more of a health system indicator. A bit more discussion would help. My interpretation is that there should be variation in quality indicators otherwise there is no performance gap or area to target quality improvement. With that said, I really liked that the authors chose two emergent conditions (AMI and hip fx) for comparison purposes. If the authors chose to make this final conclusion about the indicator, I think more information to substantiate the argument is warranted. Otherwise, their statements could be softened.

Response: Thank you for your considered feedback. In Australia the indicator is framed as an indicator of access to and quality of primary care, and indeed there are concerns around which part of the health system should be accountable. Our conclusions actually align with your suggestion that the indicator is reflective of the health system more broadly, and that both the health system context and local models of care are essential to interpretation of variation. Similar feedback was given by Reviewer 3 (Comment 7), and we have updated the abstract, discussion and conclusions to modify our conclusion accordingly.

Abstract (page 2): "Conclusions: Geographic variation in preventable hospitalisation is determined in part by hospitals, reflecting different roles played by community and multipurpose facilities, compared with major and principal referral hospitals, within the community. Care should be taken when interpreting the indicator simply as a performance measure for primary care"

Discussion (page 7-8): "The preventable hospitalisations indicator is considered a measure of timely and effective access to primary care, and our findings are not inconsistent with this interpretation. Some of the variation in community and multipurpose hospitals is likely to reflect the facility acting as a substitute for primary care in areas where access is poor, and may arguably reflect either a deficiency of primary care or appropriate integration of services to meet population needs. We were unable to examine further dimensions of access, such as waiting times, distance to nearest GP clinic and type of in-hospital practitioner, so were unable to further tease out these effects. However our results do suggest that use of the preventable hospitalisations indicator beyond its original intent—as a yardstick measure of health system performance⁷—needs to be approached with caution."

Conclusions (page 8): "Geographic variation in rates of preventable hospitalisation is determined in part by the hospitals themselves, reflecting different roles of smaller and rural hospitals compared with major and principal referral hospitals to meet the needs of the community. International adoption of the preventable hospitalisations health performance indicator should consider the contextual barriers and facilitators to accessing care in the relevant health system. In Australia, care should be taken when interpreting preventable hospitalisations simply as a measure of accessibility and quality of primary care."

Reviewer: 3

I would thank for the opportunity to review this interesting study which aims to provide new evidence to refute the idea that preventable hospitalisations (PH) can be used as a mere indicator for primary care performance.

At this respect, there is published evidence showing that PH might be associated with different factors, some of them related to primary (ambulatory) care, such as, effective access to healthcare facilities, care continuity across levels or availability of primary care professionals;

whereas others are non-ambulatory care related, for example socioeconomic variables, existence of long-term care services, and some hospital related factors such as supply of acute beds or propensity to hospitalisation.

In this work, authors specifically focus on the potential influence of hospitals in PH by means of quantifying variation in preventable hospitalisations between hospitals in order to better understand the role of those in that type of admissions.

The work is well structured and, in general, clearly explained, resulting in a smooth and interesting reading.

Response: Thank you, we are glad you enjoyed the manuscript.

Comment 7: My major objection is about the conclusion that authors inferred from their analysis

To get their objective, authors include different categories of hospitals in the study, some of them admitting non acute patients. I consider that the purpose and population served by these centres should be detailed (apart from appendix 2) to ease the understanding by non-Australian readers. Reading the text, I wonder if some community and multipurpose centres that admit non-acute patients could have been specifically devised to admit and treat patients with the conditions considered avoidable because of the lack of ambulatory facilities in the area where they are placed.

In this line, supply of GP services is the only variable used referring to primary care and maybe not sufficient to cover all the potential influence of primary (or ambulatory) care in avoidable hospitalisations. Maybe difficulties to access primary care centres or the lack of ambulatory facilities that assure care continuity, are affecting the propensity detected. Therefore, it could not be discarded the effect of inappropriate ambulatory care in the higher risks of admissions detected in some hospitals.

Response: We agree with your interpretation that some community and multipurpose hospitals may be acting as an appropriate substitute for primary care in areas where access to care is poor, and this hypothesis had been previously discussed (e.g. Introduction, paragraph 3). We have expanded on the methods to further detail the role of community and multipurpose facilities (page 5), and have further elaborated in the abstract, discussion and conclusions, and softened our conclusion accordingly (See response to Comment 6 above).

“Australia has a vast geography with most high-volume facilities located in metropolitan and inner regional areas. The smaller community and multipurpose facilities provide a mix of acute and sub-acute care, with multipurpose able to provide a range of integrated care services as negotiated between government, health practitioners and the community.”

Comment 8: If possible, other variables referring to accessibility or utilisation (distance to primary care centres) of primary/ambulatory care should be included in the analysis, or if it is not possible, it should be addressed in discussion/limitations.

It has to be said that authors qualify the results in the discussion (lines 26-30), but in conclusions they stated that PH are determined in part by hospitals, despite the fact that according to the results, the differences in admitting patients between hospitals may be a consequence of the lack of accessibility to appropriate care in some areas.

I consider this work will benefit from considering additional variables related to primary/ambulatory care and/or further discussion about the interpretation, implications and conclusions derived from the results.

Response: We have been unable to include further measures of accessibility to primary care (e.g. distance to GP clinics) as these are not readily available, or able to be derived, at the population-level in Australia. As per Comments 6 and 7 above, we have updated the abstract, discussion and conclusion to further discuss this limitation, and interpretation of our findings.

Minor things.

Comment 9: In results, authors stated “When stratified by category of hospital, the greatest variation was seen in community, district and multipurpose hospitals, [...]”. But in figure 1, only one multipurpose hospital has a significantly different risk from the mean and the ARD of this group (27.3) is similar to the ARD calculated for major metropolitan hospitals, so the sentence should be modified accordingly.

Response: We agree with the reviewer’s interpretation and have updated the sentence accordingly (page 6).

“When stratified by category of hospital, the greatest variation was seen in community and district hospitals, with community hospitals in particular having the highest rates of preventable hospitalisation”

Comment 10: Finally, there are some points in Methods section that will benefit from further explanation:

- A definition of full time workload equivalent general practitioners

Response: Further details on the calculation of full time workload equivalent GPs have now been provided (page 4).

“...and the effective supply of full-time workload equivalent (FWE) general practitioners (GPs). FWE GPs were derived from aggregated Medicare claims data,^{9,24} as the number of claims for GP services for residents of each SLA, divided by the average number of claims per FWE GP in NSW. Population estimates were used to calculate the density of FWE GPs per 10,000 residents of each SLA, and divided into quintiles.”

Comment 11: - The calculation of overall IRRs for hospital (current explanation “overall IRRs for hospital types were derived by including the hospital category in the model, as a 10% increase in provision of hospital services to the patient’s postal area, centred on the mean group value” is not explanatory enough),

Response: These methods have been further elaborated (page 5).

“Overall IRRs for hospital types were derived by adding parameters for each hospital type in the model. Given the multiple membership structure, the parameters were calculated as the proportion of hospital services provided by each hospital type in the patient’s postal-area. Each parameter was centred on the mean group value, and scaled so a single unit increase represents a 10% increase in service provision.”

Comment 12: - Detailing the number of hospitals included in each category and the number of short-stay and long-stay admissions included in the sensitive analysis.

Response: The number of hospitals in each category have now been added to the results (page 6).

“Patients were admitted to a total of 259 different facilities, including n=17 principal referral, n=12 major metropolitan, n=12 major non-metropolitan, n=38 district, n=70 community and n=110 multi-disciplinary facilities.”

The number of short- and long-stay admissions have now been added to the results (page 7).

“A sensitivity analysis categorising length of stay (Table 3) found more the majority of preventable hospitalisations (n=16,305, 53.9%) were short stay admissions (0-2 day LOS), with the remainder (n=13,959, 46.1%) having a LOS of three days or more. There were differing patterns of variation...”

FORMATTING AMENDMENTS (if any)

Required amendments will be listed here; please include these changes in your revised version:

Comment 13: Please provide better qualities figures, ensuring the figures are not pixelated when zoomed in on. Figures can be supplied in TIFF, JPG or PDF format (figures in DOCUMENT, EXCEL or POWERPOINT format will not be accepted), we also request that they have a resolution of at least 300 dpi and 90mm x 90mm of width. *figure uploaded only 72 dpi, should be at least 300 dpi

Response: A higher resolution figure, at 300 dpi, have been uploaded accordingly.

Comment 14: Patient and Public Involvement: We have implemented an additional requirement to all articles to include 'Patient and Public Involvement' statement within the main text of your main document. Please refer below for more information regarding this new instruction:

Authors must include a statement in the methods section of the manuscript under the sub-heading 'Patient and Public Involvement'.

This should provide a brief response to the following questions:

How was the development of the research question and outcome measures informed by patients' priorities, experience, and preferences?

How did you involve patients in the design of this study?

Were patients involved in the recruitment to and conduct of the study?

How will the results be disseminated to study participants?

For randomised controlled trials, was the burden of the intervention assessed by patients themselves?

Patient advisers should also be thanked in the contributorship statement/acknowledgements.

If patients and or public were not involved please state this.

Response: A new sub-section has been added to the methods accordingly (page 5). Patients and the public were not involved in the design of this study.

“Patient and public involvement

Participants in the 45 and Up Study completed a baseline questionnaire and have provided informed consent for the use of their data for research purposes. However, patients and the public were not involved in the design of this study.”

VERSION 2 – REVIEW

REVIEWER	Reviewer name: Matthew Hensley Institution and Country: University of Michigan, United States of America Competing interests: none
REVIEW RETURNED	17-Dec-2018

GENERAL COMMENTS	Falster et al. conducted an investigation testing whether there is between-hospital variation in preventable hospitalizations using a multi-level, multi-member Poisson regression. This observational study analyzed a group of patients >45 years in New South Wales, Australia. Overall the methods were thoughtful and statistical methods appropriate. The authors had access to a large cohort with survey data, which is a clear strength of the study. The population was older and may not be generalizable to the entire Australian population, however the results demonstrate significant between-hospital variation with regards to preventable hospitalizations, after adjustment for multiple factors. Given the cohort was population-based, complete identification of comorbidities using an index such as Charlson or Elixhauser, could not be fully assessed, which may have impacted results. The study's results have important implications for health policy in Australia. The conclusions do not support relying solely on preventable hospitalizations as a measure of primary care quality/access.
---

REVIEWER	Reviewer name: Laurie Coats Daras, PhD Institution and Country: RTI International Competing interests: None declared
REVIEW RETURNED	17-Dec-2018

GENERAL COMMENTS	The authors sufficiently addressed my prior comments in their revision.
---

REVIEWER	Reviewer name: Ester Angulo-Pueyo Institution and Country: Aragon Health Sciences Institute (IACS) Competing interests: None declared
REVIEW RETURNED	20-Dec-2018

GENERAL COMMENTS	Authors have addressed satisfactorily all the comments and suggestions I have made in the previous revision. My recommendation is Accept the manuscript.
--